# An Assessment of the Effect of Progressive Water Absorption on the Interlaminar Strength of Unidirectional Carbon/Epoxy Composites Using Acoustic Emission

**DOI:** 10.3390/s21134351

**Published:** 2021-06-25

**Authors:** Faisel Almudaihesh, Stephen Grigg, Karen Holford, Rhys Pullin, Mark Eaton

**Affiliations:** Cardiff School of Engineering, Cardiff University, Queen’s Building, The Parade, Cardiff CF24 3AA, UK; s.grigg35@gmail.com (S.G.); Holford@cardiff.ac.uk (K.H.); pullinr@cardiff.ac.uk (R.P.); eatonm@cardiff.ac.uk (M.E.)

**Keywords:** water absorption, carbon-epoxy, interlaminar damage mechanism, acoustic emission, optical measurement, microscopy

## Abstract

Carbon Fibre-Reinforced Polymers (CFRPs) in aerospace applications are expected to operate in moist environments where carbon fibres have high resistance to water absorption; however, polymers do not. To develop a truly optimised structure, it is important to understand this degradation process. This study aims to expand the understanding of the role of water absorption on fibrous/polymeric structures, particularly in a matrix-dominant property, namely interlaminar strength. This work used Acoustic Emission (AE), which could be integrated into any Structural Health Monitoring System for aerospace applications, optical strain measurements, and microscopy to provide an assessment of the gradual change in failure mechanisms due to the degradation of a polymer’s structure with increasing water absorption. CFRP specimens were immersed in purified water and kept at a constant temperature of 90 °C for 3, 9, 24 and 43 days. The resulting interlaminar strength was investigated through short-beam strength (SBS) testing. The SBS values decreased as immersion times were increased; the decrease was significant at longer immersion times (up to 24.47%). Failures evolved with increased immersion times, leading to a greater number of delaminations and more intralaminar cracking. Failure modes, such as crushing and multiple delaminations, were observed at longer immersion times, particularly after 24 and 43 days, where a pure interlaminar shear failure did not occur. The observed transition in failure mechanism showed that failure of aged specimens was triggered by a crushing of the upper surface plies leading to progressive delamination at multiple ply interfaces in the upper half of the specimen. The crushing occurred at a load below that required to initiate a pure shear failure and hence represents an under prediction of the true SBS of the sample. This is a common test used to assess environmental degradation of composites and these results show that conservative knockdown factors may be used in design. AE was able to distinguish different material behaviours prior to final fracture for unaged and aged specimens suggesting that it can be integrated into an aerospace asset management system. AE results were validated using optical measurements and microscopy.

## 1. Introduction

Carbon fibre reinforced polymer composites are used in many industries due to their excellent fatigue and corrosion-resistant properties as well as their high strength-to-weight and stiffness-to-weight ratios [1]. However, the performance of CFRP composites can be limited in real-life applications due to degradation or ageing caused by processes that take place in a moist or harsh environments, particularly within polymers [2]. Many structures made of composite materials are used in the aerospace industry, including wings, ailerons, and fuselages of aircrafts. They are also used in the main and tail rotor blades of helicopters. These structures are constantly exposed to moist environments. The presence of moisture in large composite structures is monitored by NDT methods such as the infrared thermography [3]. Quantifying moisture within large composite structures is essential to monitor the progress of water ingress but it can be rather difficult to quantify [4]. However, in small structures (small test specimens), there are well-established standards that allow quantification of moisture in composite samples using gravimetric methods to monitor mass change [5]. This is then followed by mechanical testing to determine the influence of moisture absorption on the mechanical properties [6,7,8] and therefore allows better understanding of the behaviour of the materials at certain moisture absorption levels, leading to safer structural designs. The absorbed water can cause reversible and/or irreversible changes in the physical and chemical properties of the material [7,9,10,11,12,13,14] because polymers in composite materials chemically react with water since epoxy resins have the ability to absorb substantial amounts of moisture. These reactions are triggered by the polar groups in the material and are caused by the replacement of ester groups with alcohol and carboxyl acids that attract water molecules, leading to plasticisation [9]. Plasticisation is the swelling of the membranous structure of polymers due to sorption of a penetrant in a polymer matrix. The penetrant can be either carbon dioxide or water in addition to a range of linear hydrocarbons and aromatics. This process generally yields an increase in the fractional free volume of the membrane, resulting in the diffusion of all the gas species via the membrane. This is followed by an increase in permeability and loss of selectivity. For example, the plasticisation of BTEX aromatics at 200–1000 ppm can reduce the membrane’s selectivity by 30–50% [15]. However, the ageing of membranes is essentially the reverse of plasticization—compaction of membranous structures over time, resulting in loss of fractional free volume. This results in loss of permeability with increased selectivity [15]. Plasticisation effects also decrease the glass transition temperature (Tg) of composite materials [16]. These physico-chemical changes affect both diffusion and hydrolysis kinetics. It is the hydrolysis of the reactive groups, present in a polymer’s structure, that causes the detrimental damage. Hydrolysis causes chain scission, which decreases a polymers’ crosslink density [9]. Moreover, chemical scissions due to hydrolytic degradation occur on the same elastically active chain. This results in relatively low molecular weight species that have the tendency to leave the polymer network through lixiviation [9,17]. This process results not only in global degradation but also in chemical and structural changes that turn into a complicated diffusion-reaction coupled phenomena [9,17]. The absorption processes of epoxy resins can be complicated and often reveal various anomalies in the mass uptake. It has been found that part of the absorbed water is present as free molecules in the micro-voids of the network; alternatively, it is involved in physical interactions with the polymer chains or other water molecules [18]. The rest of the water molecules are involved in hydrolysis reactions within the material [17]. Therefore, the total absorbed water cannot be considered an indication of a unique ageing process since it has different effects on the polymeric constituent. It is necessary to examine the absorption of moisture molecules, with the subsequent evolution of properties, in terms of swelling. For this purpose, spectroscopy techniques, such as the Nuclear Magnetic Resonance (NMR) and Fourier Transform Infrared (FTIR) are used [19]. Generally, water absorbed into composite materials after their immersion in water is categorised as two types: Free water and bound water. Free water is transported into a polymer via microcracks, whereas bound water is dispersed in the polymer matrix and attaches to the polymer’s polar groups [20]. Knowledge of the mechanisms that drive moisture absorption, as well as the influence of dimensions, temperature, and relative humidity, is important when the long-term properties of the composite are of interest [21]. It is crucial to understand the three common methods of moisture absorption by polymeric composite materials, namely the diffusion process, capillary actions, and the transport of water molecules. The diffusion process takes place inside the micro gaps between polymer chains [22,23]. The capillary process occurs in the empty spaces between the fibre-matrix interfaces. These spaces within the interfaces are produced during the manufacturing process, particularly when the reinforcement’s impregnation into the matrix is incomplete. The transport of water through microcracks is the most important mechanism that leads to the ultimate debonding of the fibre-matrix [13,24,25].

The study of the interaction of polymers with water at the molecular level is very important because water intake determines a material’s mechanical properties at the macroscopic level [21] and any moisture absorption by polymeric composite materials can result in permanent damage to the matrix and interface [19,26]. This ingress of water molecules is significant if the polymer-based structure is immersed in water, salt, or alkaline-based solutions because this leads to a reduction in stiffness, strength, and the resin’s glass transition temperature (Tg) due to plasticisation effects [2,7,10,11,12,13].

Interlaminar shear strength is often a limiting factor of fibrous-polymeric composites [27]. The in-plane properties of the laminate depend primarily on the fibres whereas the out-of-plane (through the thickness) properties such as SBS, historically referred to as interlaminar shear strength (ILSS), depend largely on the polymeric constituents of the composite [28]. The interfacial fibre-matrix properties are also a key factor due to the fact that load on the composite is transferred from the matrix to the fibres through the interface; hence fibre-matrix interfacial shear strength can affect the toughness and strength of the composite [29,30]. Therefore, moisture conditions will most likely result in changes on the mechanical properties of the matrix and/or the interfacial properties between the fibres and the matrix [31]. A consistent and significant loss of SBS is seen in CFRP specimens, with reductions of up to 21% after 20 weeks of immersion in distilled water at 25 °C [32]. Meng et al. immersed [±45]_4s_ and [90/0]_4s_ CFRP specimens in tap water at 50 °C for one month. The [±45]_4s_ specimens showed an increase in cracks presence but little SBS variation between un-aged and aged specimens, whereas the [90/0]_4s_ specimens showed a 20% reduction in SBS after one month [33]. Barbosa et al. [7] reported that there was a drop in CFRP composites’ SBS values after water absorption and the aged specimens showed a sharp drop in applied load from the load-versus-displacement graphs, followed by its further increase until the final failure is observed. The authors suggested that this behaviour is due to the potential formation of microcracks in the composites. This study further investigates this behaviour and the evolution of gradual change in the interlaminar strength of CFRP composites following water immersion for increasing exposure time, focussed upon the examination of a specimen’s load history, representative AE features, and in-test fracture imaging, allowing a detailed understanding of the change in failure behaviour that occur.

The AE technique has been applied to composites in order to better understand their damage behaviour and evolution [34]. It is defined as the spontaneous release of elastic energy during the growth of damage. This energy propagates through structures as a transient wave [35] and can be detected by piezoelectric sensors [36]. From these recorded waveforms, the parameters summarised in Figure 1 can be extracted [37]. Many researchers have attributed various AE parameters to the characteristics of composite materials in order to gain deeper insight into the damage mechanism [36,38,39,40,41]. Kumosa et al. [41] reported that AE can be used to distinguish between different types of damage in glass-fibre-reinforced polymers (GFRP). They state that high-amplitude events are associated with fibre fractures, whereas low-amplitude events are associated with matrix cracking. De Groot et al. [40] used AE frequency distribution to identify damage mechanisms in CFRP. Pérez-Pacheco et al. [12] used the AE (duration versus amplitude) under tensile tests to study CFRP composite behaviour after the fibre/matrix interface was exposed to moisture. The authors correlated the physical failure mechanism with the AE response (duration versus amplitude), as shown in Table 1.

This paper studies the evolution of damage mechanisms in CFRP material following water immersion for increasing durations. It focuses on the matrix-dominated property of interlaminar strength; additionally, examination of load history, in-test fracture imaging, and representative AE features provided a detailed understanding of the change in failure behaviours that occur. Therefore, this work reflects on the understanding of the effects of gradual water absorption on the delamination resistance of fibrous/polymeric composites. It also explains, from a micro-scale perspective, the fracture mechanism observed prior to final fracture and provides an overall evaluation of the SBS testing method with the use of non-destructive test (NDT) techniques. This allows better performance predictions of the behaviour of CFRP structures, producing safer application designs for operating in moist conditions such as in aerospace, automotive, and energy.

## 2. Materials and Methods

### 2.1. Materials

A 15k unidirectional (UD) Pyrofil TR50S high-strength carbon fibre 200 gsm with 33% Skyflex K51 epoxy resin content (the glass transition temperatures (tg): From Dynamic Mechanical Analysis (DMA) in accordance with ASTM D7028 [42], is 122.83 °C) was used for this study. The materials were cured as recommended by the supplier, using an autoclave under two dwells for 30 min at 80 °C at five bar pressure followed by 60 min at 125 °C at five bar pressure. Large panels, consisting of 30 plies (5.77 mm thick) with a layup of [0/90]15, were prepared. Due to the sample thickness, a debulking operation (5 min vacuum) was undertaken after each five plies [43]. The stacking sequence was implemented to simplify the damage mechanism by excluding additional fibre directions and to avoid symmetric plies in the mid plane. All panels were inspected through c-scanning (MIDAS NDT Systems Ltd, Ross-on-Wye, UK. with pulse-receive mode using a 10 MHz probe) to ensure manufacturing quality.

### 2.2. Water Absorption

Water absorption testing was carried out using non-ambient moisture conditioning in a water immersion tank at a prescribed constant temperature of 90 °C for 3, 9, 24, and 43 days. Purified water (type 1) was used in an unstirred digital bath (NE2-28D), supplied by Clifton, with a sensitivity of ±0.2 °C and uniformity of ±0.1 °C. Specimens were placed on a support rack and immersed in the bath once a steady-state temperature was reached. To record specimen mass, specimens were individually removed from the water bath and left in a sealed bag until an acceptable handling temperature (room temperature) was reached. The specimens were then removed from the bag, the surface moisture was wiped off, and the mass change was measured using an analytical balance with an accuracy of 0.0001 g (Sartorius LA310S analytical balance. Sartorius AG, Göttingen, Germany). The water absorption test setup is in accordance with ASTM D 5229 [5]; however, a higher water temperature was chosen to accelerate the ageing process and monitor the progressive damage mechanism during ageing. In order to obtain a realistic prediction of long-term service behaviour, it is suggested that the immersion temperature should be at least 20 °C lower than the tg of the matrix [44]. At 90 °C the specimens considered here were ≈33 °C below their tg based on E’ onset from DMA measurements. This avoids secondary degradation mechanisms (excessive, unwanted degradation) activated at temperatures exceeding the tg [45,46]. The water type used for this study excludes any potential chemical reaction between the polymers and the natural containments that might exist in other types of water. The specimen water content was determined as a percentage change using Equation (1) [5].
(1)$$\mathrm{Mass}\;\mathrm{percentage}\;\mathrm{change},\;\%\;=\left(\frac{W_{i}-{\;W}_{o}}{W_{o}}\right)\times 100$$
where:W_i_ = weight of the specimen at each point of the weight recorded during the experiment.W_o_ = initial weight of the specimen before any contact with water.

### 2.3. Interlaminar Strength

The interlaminar strength was observed by applying the SBS test, using a three-point bend setup. The diameter of the support rollers is 3 mm, the diameter of the loading nose is 6 mm, and the span is 24 mm. A Zwick Roell Z050, ZwickRoell Ltd, Leominster, UK load machine with a 50 kN load cell was used to apply a load at a crosshead rate of 1 mm/min, in accordance with ASTM D2344 [47]. The SBS was calculated using Equation (2) [47].
(2)$$F_{\mathrm{sbs}}=0.75\times \frac{P}{b\times h}$$
where:F_SBS_ = short-beam strength (MPa).P = load observed during the test (N).b = specimen width (mm).h = specimen thickness (mm).

The testing matrix for this study is shown in Table 2.

### 2.4. Acoustic Emission

AE signals were recorded during each static test using a MISTRAS PCI2 AE system, Mistras Group Ltd, Bridgend, UK. The same PICO (MISTRAS) sensor was used for each specimen and bonded with superglue in the same location, as shown in Figure 2. The sensor type was chosen due to the specimen’s small dimension. The operating frequency range for the PICO sensor is 200–750 kHz, and the resonant frequency is 250 kHz. The sensor was mounted to the top surface of the specimen using superglue, which provides physical attachment and acoustic coupling. A MISTRAS 2/4/6 preamplifier with gain set to 40 dB was used with an analogue filter between 20 kHz and 1200 kHz. The AE system detection threshold was set to 45 dB. The peak detection time (PDT), hit definition time (HDT), hit lockout time (HLT), and max duration were 200 µs, 800 µs, 100 µs, and 100 ms, respectively. The progressive growth of damage was monitored by acquiring four parameters from the AE waveforms, which were sampled at 40 MHz, these parameters were absolute energy, amplitude, duration, and counts (Figure 1). Both load and displacement were also recorded at a rate of 10 Hz and with each hit.

### 2.5. Optical Measurements

An Imetrum Video Strain Gauge (VSG) system was used for this study. A 5 MP IMT-CAM028 camera, Imetrum ltd, Bristol, UK with IMT-LENS-MT043 material test lenses was used with a field of view of 40 × 36 mm and a record rate of two frames per second. Both load and displacement from the testing were recorded with each image. A speckle pattern was applied to all specimens for subsequent video strain gauge analysis. This was achieved by spraying the specimens with a light dusting of matt white spray paint (a light layer so that it does not build a thick layer that would eliminate characterising the specimens actual surface) followed by a light dusting of matt black paint, in line with Imetrum guidance [48].

### 2.6. Microscopy

A Leica DM LM microscope, Leica Microsystems Ltd, Milton Keynes, UK was also used for this study. The images were then exported using OmniMet software attached to the microscope.

## 3. Results and Discussion

Figure 3 shows the mass gain resulting from water immersion and the consequential reduction in SBS (error bars represent the standard deviation). The mass change follows an expected trend of weight gain until a saturation state is reached after approximately 35 days, and the SBS is seen to correspondingly reduce with the increase of immersion times [7,14]. Carbon fibres in CFRP composites are very resistant to water swelling and do not absorb water [49]; thus, the reduction in SBS is believed to be due to plasticisation of the matrix, along with microcracking and interfacial degradation, induced by the moisture absorption process in the aged specimens, which, therefore, reduces the fibre supports, further demonstrating the role of polymers within the composite structure after aging [46,49,50,51,52,53,54]. The water uptake in fibrous composites can be surmised in three stages; stage (I) is where moisture uptake and material decomposition are competing until the condition of moisture saturation is reached [14]. During this stage, the moisture uptake is greater than the mass loss (can be reversed by drying if the water molecules are only filling the matrix microcracks and not yet reacted with the polymers). In stage (II), moisture uptake has reached a maximum, or its rate of increase is equal to or less than the rate of mass loss into the immersion solution [14]. In stage (III), the decomposition rate is dominant (well into the irreversible changes) [14]. The irreversible changes mainly take place when the polymers chemically react with water. This leads to hydrolysis, which has a plasticising effect leading to a reduction in properties [9,10,12,17].

Figure 4 shows the SBS versus displacement curves for all specimens. As observed in the literature, two expected behaviours are seen: Higher strain at lower SBS for aged specimens and a drop in the applied load from the load versus displacement followed by its further increase until the final failure. To further understand these two known findings, AE, optical measurements, and microscopy are used for the analysis.

In (Figure 4), the modulus is seen to be reduced, and the curves become more non-linear (evidence of yielding in the matrix) as immersion time increases. As the carbon fibres are not affected by moisture, this confirms the plasticisation of the epoxy matrix due to moisture ingress, as expected. Figure 5 shows 2D DIC strain maps of a representative specimen at 40 MPa SBS. The colour contours represent shear strain, and a symmetric behaviour is seen in the specimen, with higher strains observed at the mid-plane. The shear strain is seen to increase with immersion time as the matrix properties degrade, which corresponds to the behaviour seen in the SBS versus deflection curve (Figure 4). Figure 6 shows the cumulative counts from the AE data versus time (one representative specimen is shown for each aging category, and details for all remaining specimens are attached in the Appendix A). The higher shear strain seen in aged specimens (Figure 5) is believed to have led to more crack formation at this point, which is demonstrated by higher AE cumulative counts (Figure 6).

The sharp drop in the applied load from the load versus displacement, followed by its further increase until the final failure, as shown in section A of Figure 4, is in agreement with the findings of Barbosa et al. [7], who correlated this behaviour to the potential formation of microcracks in the composites. For this study, this behaviour was investigated further in order to understand any influence on failure mechanics and therefore on the interlaminar properties determined.

Figure 7 shows the AE absolute energy versus time for all specimens with a maximum absolute-energy scale bar of 1.0 × 10^7^ in order to show the significant AE signals associated with high energy, which increases with an increase in water immersion time, and this signal also correlates well with the drop in applied load versus time prior to final fracture, which suggests that there are significant damage mechanisms taking place at this point (one representative specimen is shown for each aging category, and details for all remaining specimens are attached in the Appendix A). Figure 8 is similar to Figure 7, but with a maximum absolute-energy scale bar that shows all absolute energy signals detected prior to final fracture. The drop in applied load and its association with the high absolute energy observed in aged specimens is clearly seen in (Figure 8), particularly in the 24-day and 43-day aged specimens.

From the AE absolute energy data, it is clear that there are significant AE signals that correspond to the drop in applied load prior to final failure. To understand this crack behaviour, the amplitude and duration of the AE signals observed from these specimens were used to investigate the physical damage mechanism associated with this behaviour. In order to do this, the findings in (Table 1), taken from the study by Pérez-Pacheco et al. [12], were used. Their study correlated duration versus amplitude to the physical damage of CFRP. However, that study was conducted on tensile specimens, whereas the tests here were conducted under bending (SBS). To use their data as a reference, it is assumed that the AE activity released from fractures in composite samples is similar regardless of the load case. For instance, the release of AE activity when fibres break in a tensile test is similar in a shear test because in both cases, reference is made to the energy from the fibre breakage. This is not necessarily an accurate assumption because a complex failure mechanism is expected under shear tests [55]. Although the assignment of damage mechanisms to each class cannot be applied with confidence to this testing, it does provide a framework for classification from which comparisons can be made between samples. Recorded signals for each test have been classified using the ranges given in (Table 1) and the occurrence of signals from each class are plotted cumulatively against the time of test in Figure 9 (one representative specimen is shown for each aging category, and details for all remaining specimens are attached in the Appendix A). The occurrence of signals classified as class 1 is seen throughout all tests with a significant increase observed towards the end of the test. In the case of the unaged and 3-day aged specimens, very little activity is seen from other classes. In samples aged for 9 or more days, other classes become more evident and their occurrence seems to be correlated to the load drops observed prior to final failure.

To investigate this change in damage behaviour, images taken from the VSG system and microscope images were used in order to visualise the damage in all aging groups and associate the groups with the observed AE data (a video clip for all fractures is attached in the Appendix A, and the small fractures under the load nose are really observable in the video along with crack formation and propagations for all specimens).

For the dry (control) specimen, crushing in the upper 0° ply under the loading nose after 59 s is observed, as shown in Figure 10. The post-test microscopy images shown in Figure 11, confirms and shows that no further propagation of this damage occurred, and the applied load rose until the final failure of single delamination occurred (as expected [7,26]). This shows that some damage has occurred prior to the final failure, which has led to the observed AE activity discussed above, but it is not significant enough to cause an observable drop in the load versus displacement/time curve.

After three and nine days of water immersion, similar crushing of the upper 0° ply under the loading nose is seen after 65 and 64 s, as shown in Figure 12 and Figure 13, respectively. The post-test micrographs shown in Figure 14 for the three-day aged specimen and in Figure 15 for the nine-day aged specimen show that this damage propagated further in the sample compared to the dry specimen, resulting in crushing of the second 0° ply. This local damage did not propagate any further and a final failure of a single delamination still occurred. Therefore, the drops in applied load (observed prior to final failure) are believed to correlate to the crushing behaviour along with the compressive loads local to the loading nose, which is also associated with higher shear strain near the upper 0° ply with the same SBS (Figure 5) and increased cumulative counts observed at this drop in applied load versus displacement/time (Figure 6), as compared with their dry equivalent.

At longer exposure times (24 and 43 days), crushing is also seen directly under the loading nose at the upper 0° ply after 60 s and 58 s after 24 and 43 days of immersion, as seen in Figure 16 and Figure 17, respectively. There is some noticeably high AE activity at the point of crushing in the observed data (Figure 6, Figure 7 and Figure 9), which is well correlated with the drop in the applied load prior to the final failure. The post-test micrographs shown in Figure 18 for the 24-day aged specimen and in Figure 19 for the 43-day aged specimen confirm the significant development of the crack propagation of this damage occurred at the upper 0° ply, leading to the final failure of multiple transverse cracks and delaminations occurring throughout the thickness, but predominantly in the upper half of the specimens.

In the cases of the dry, 3-day, and 9-day aged specimens, surface crushing does not appear to influence the final failure, which is by a single delamination at approximately the mid-plane. However, at longer exposure times of 24 and 43 days, surface crushing seems to be more important to the initiation of the final failure. It is suggested that this crushing leads to a buckling/peel up delamination at the first 90/0 interface, which transmits stresses into the second 0° ply from the surface, resulting in further crushing and therefore further buckling/peel up at the next 90/0 interface. This repeats every two plies at the 90/0 interface until approximately the mid-plane of the specimen. The thickness of the material between the delaminations, shown in Figure 16 and Figure 17, is ≈0.4 mm, which corresponds to two-ply thicknesses. Further evidence of this behaviour is seen in the optical microscope images (Figure 18 and Figure 19). Evidence of kink bands, commonly associated with in-plane compressive failure, can also be seen in these images. This suggests that the axial bending stresses could also be contributing to this failure behaviour. Given the larger deflections (therefore larger bending stresses) for a given SBS, seen in the 24- and 43-day aged specimens (Figure 4), this seems to be a reasonable possibility. The compressive strength of this material has also been shown to be reduced by ≈15% in an aged condition [26], further increasing the likelihood of compressive failure due to these bending stresses. This is an important observation because the later tests (24- and 43-day aged) do not represent an interlaminar shear failure; therefore, they are not a true measure of the residual short beam shear strength in the aged specimens. Rather, they are indicative of a separate and more complex failure mode brought about by the local stresses induced by the loading nose. So, while the apparent SBS is seen to be reduced with increased immersion time, it is not clear if this is actually the case. It also indicates that some of the large reductions in SBS reported in the literature (>20%) could potentially be an overestimation of the effect of moisture absorption. Other flexural tests associated with shear could be investigated, such as the four-point bend test [55] or evaluation of the suggestions by Adams et al. [56] who propose modifying the SBS test by increasing the loading nose diameter (yet to be standardised). The validation of these tests, in terms of fracture behaviour, could perhaps be done with the same or similar methods used for this study in order to draw conclusions on the reductions in shear values after aging the polymers in CFRP composites.

## 4. Conclusions

Water absorption by polymers can cause irreversible changes in the physical and chemical properties of the material. This leads to substantial changes in the overall performance of fibrous/polymeric composites. This study investigated the effects of gradual change in the status of aged polymers on the overall performance of CFRP composites. The effects are noticeable at an early stage of accelerated ageing, where the SBS is reduced by 2.8%, 11.14%, 18.78%, and 24.47% after 3, 9, 24, and 43 days of water immersion, respectively, which correlates well with observed mass changes. Higher shear strains at lower SBS values are clearly seen in specimens with longer immersion times. A corresponding increase in AE activity (presented in cumulative counts, absolute energy, amplitude, and durations) is also observed. From the literature, the drop in the applied load prior to the final fracture in aged specimens is associated with crack formation in the matrix due to water absorption. In this study, this behaviour was further investigated, and it is found that this behaviour is more complex; it is associated with crushing in the upper surface of the specimens in contact with the loading nose, leading to crack formation and propagation with more severe damage mechanisms taking place in aged specimens. It is crucial to emphasise that this change in behaviour in the overall fracture is due to physical changes taking place within the polymers due to water absorption. At longer exposure times, crushing of the upper surface occurred at a load below that required to initiate a pure interlaminar shear failure (seen as a single delamination in the control specimens), and therefore the true SBS was not determined. In conclusion, the AE, VSG, and microscopy data support the need for further research in order to develop testing standards for the evaluation of the shear properties of CFRP composites, particularly after environmental aging. This is of particular importance since most CFRP structures are expected to operate in moist conditions.

## Figures and Tables

**Figure 1 sensors-21-04351-f001:**
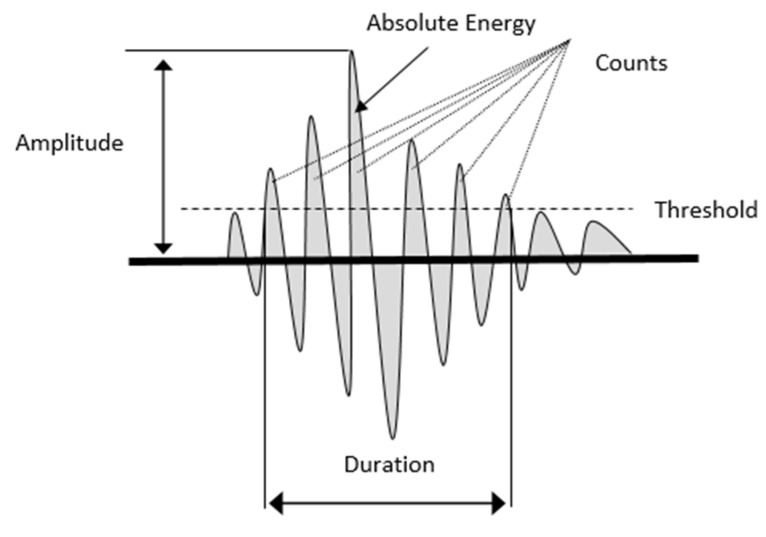
AE waveform features.

**Figure 2 sensors-21-04351-f002:**
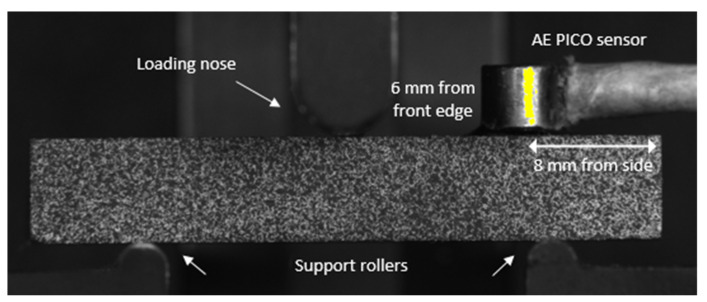
Short beam and AE test setup.

**Figure 3 sensors-21-04351-f003:**
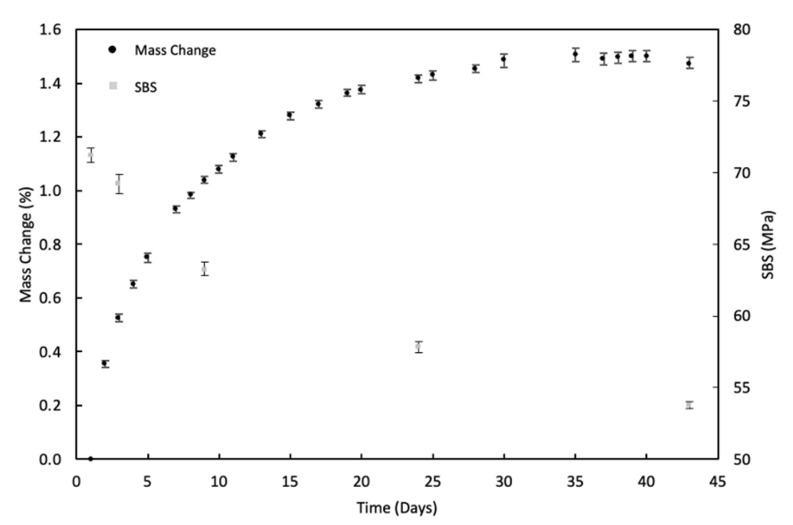
Water absorption versus SBS.

**Figure 4 sensors-21-04351-f004:**
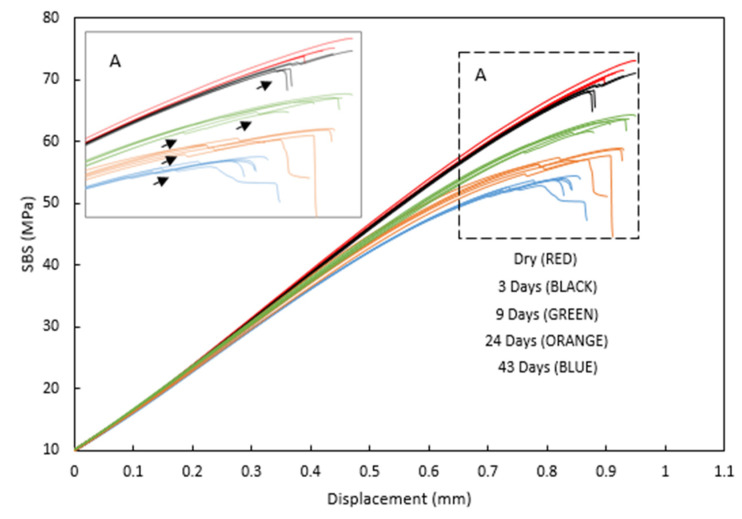
SBS versus displacement curves for all specimens.

**Figure 5 sensors-21-04351-f005:**
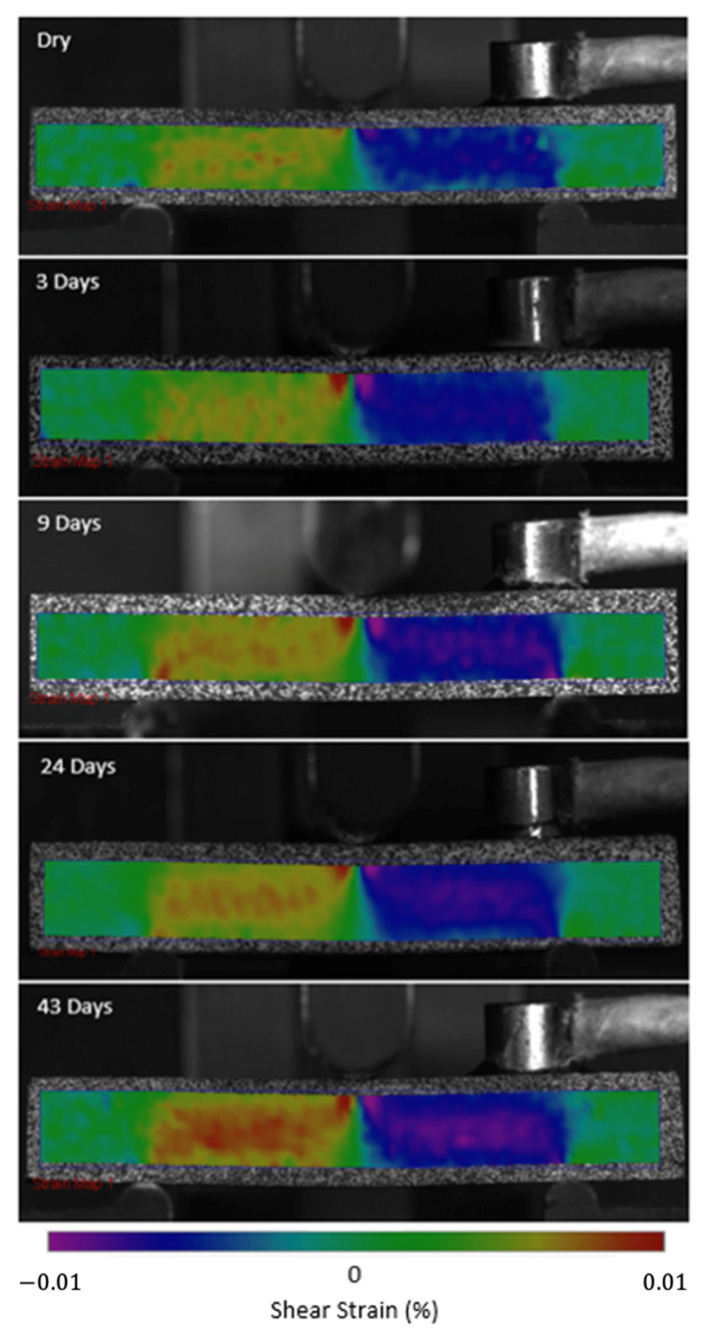
2D shear strain maps observed at 40 MPa SBS.

**Figure 6 sensors-21-04351-f006:**
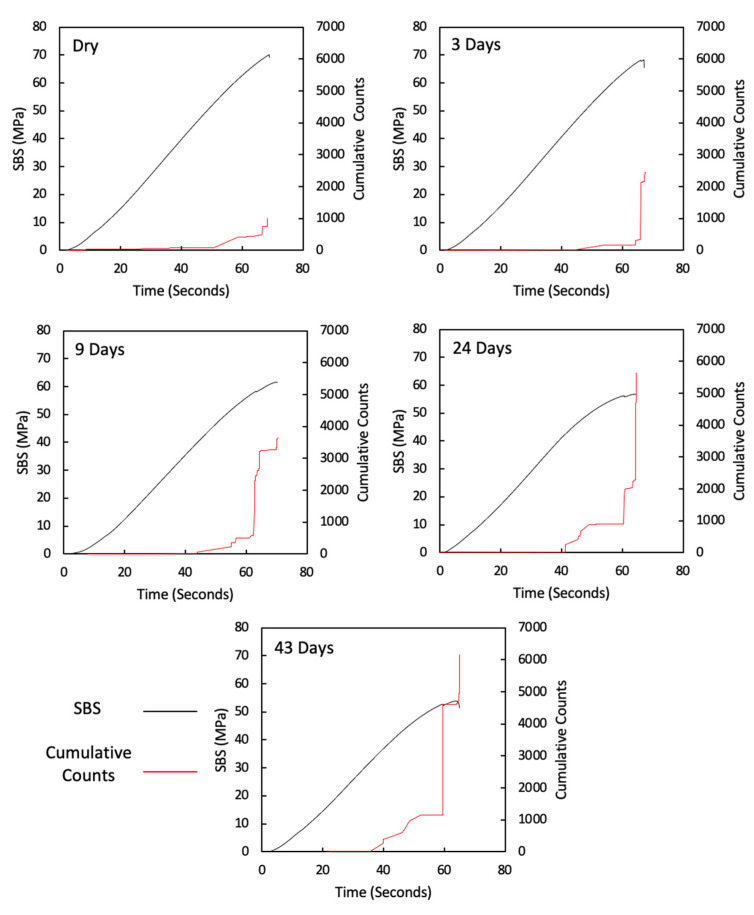
AE cumulative counts versus time for dry, 3-day, 9-day, 24-day, and 43-day water-immersed specimens.

**Figure 7 sensors-21-04351-f007:**
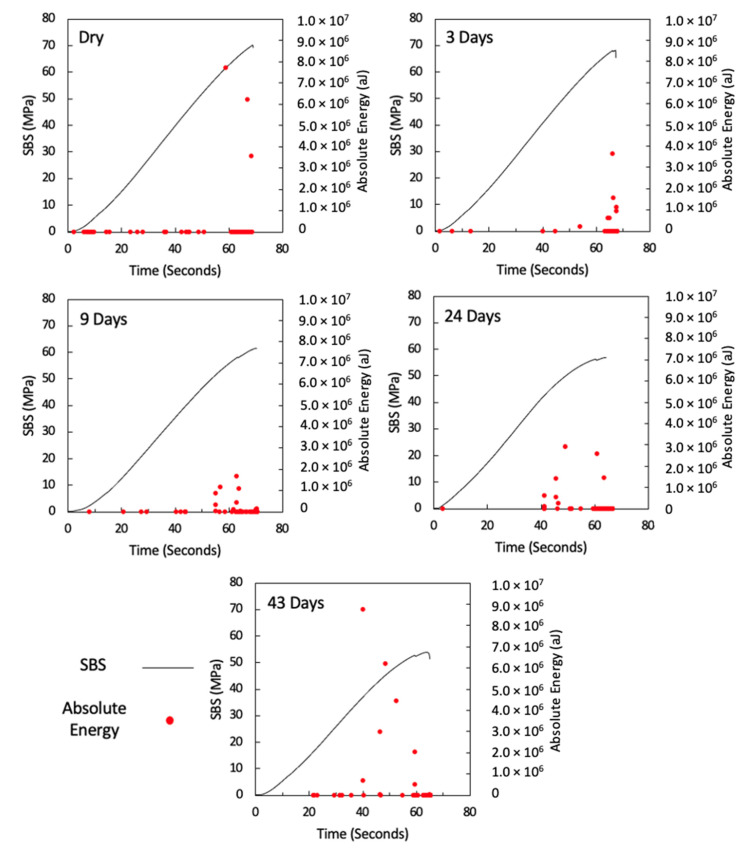
AE absolute energy versus time for dry, 3-day, 9-day, 24-day, and 43-day water-immersed specimens with an absolute energy maximum scale of 1.0 × 10^7^.

**Figure 8 sensors-21-04351-f008:**
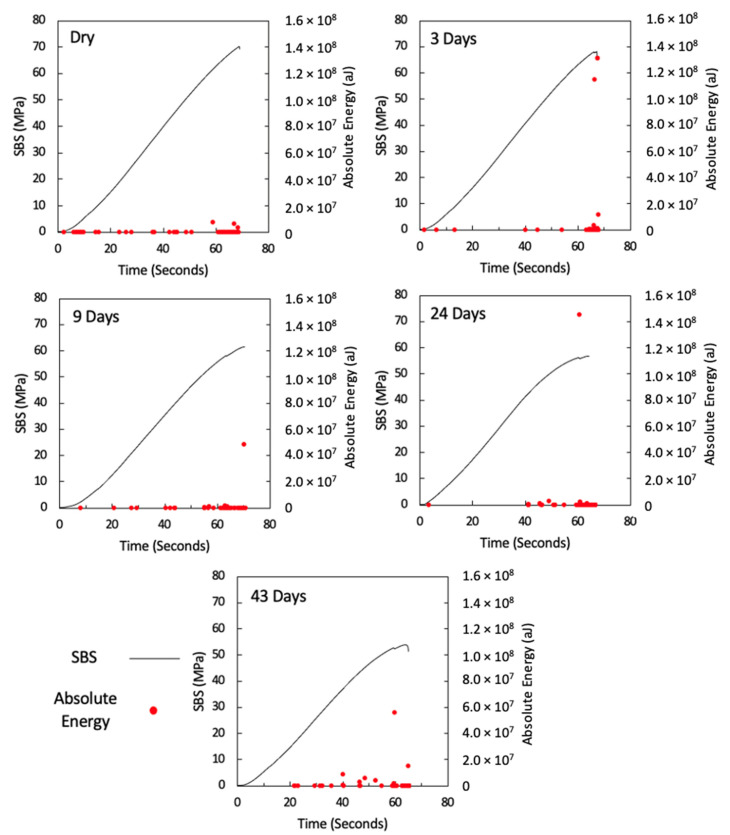
AE absolute energy versus time for dry, 3-day, 9-day, 24-day, and 43-day water-immersed specimens with an absolute energy maximum scale of 1.6 × 10^8^.

**Figure 9 sensors-21-04351-f009:**
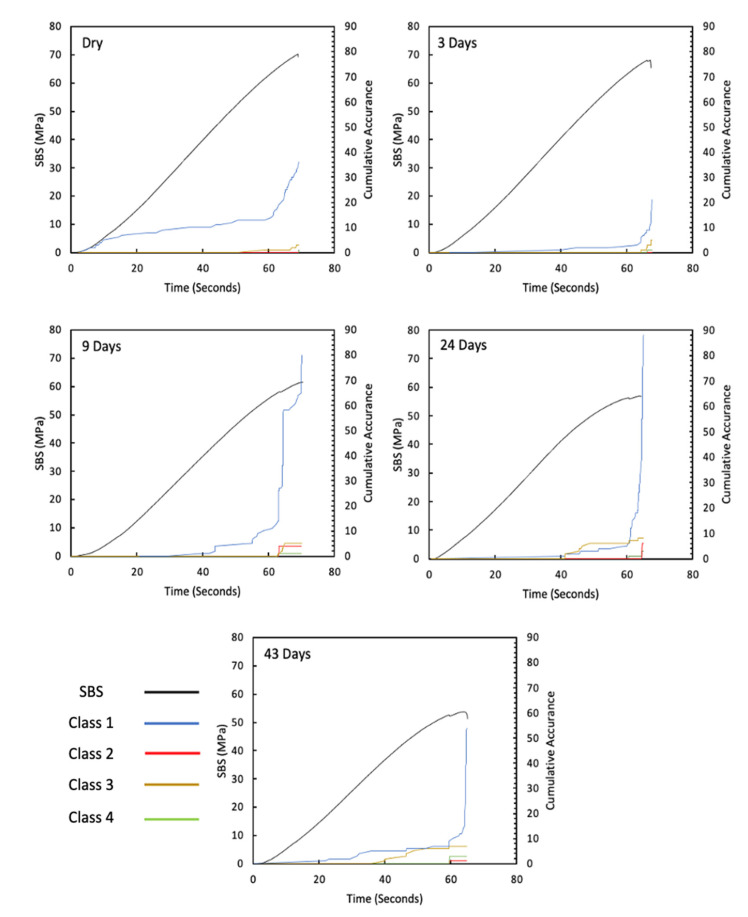
The representative damage mechanism based on the observed amplitude and duration data from the AE signals.

**Figure 10 sensors-21-04351-f010:**
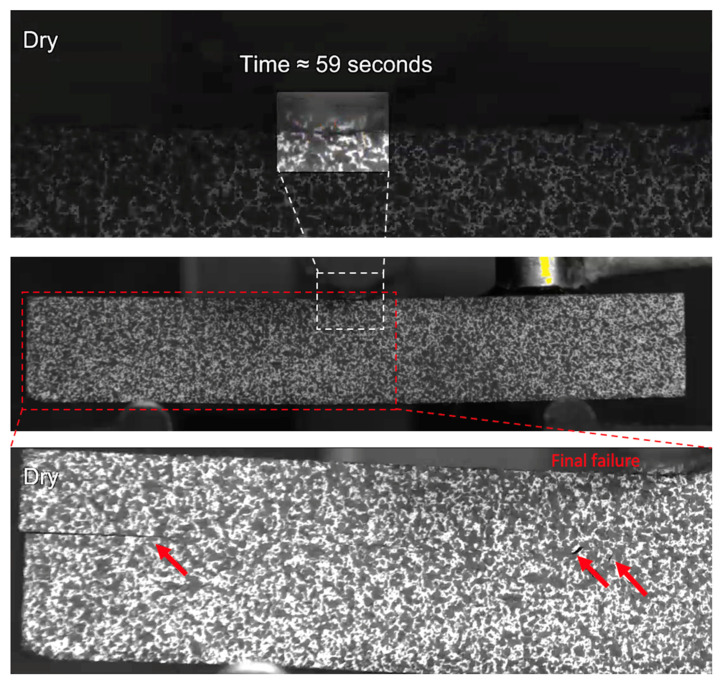
The damage observed in a dry specimen prior to final fracture. The final fracture is also seen.

**Figure 11 sensors-21-04351-f011:**
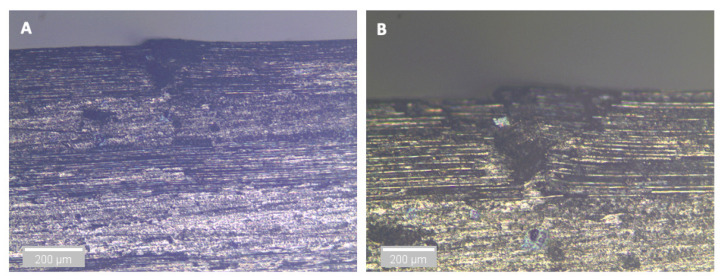
Optical microscopy of the region in contact with the loading nose for a dry specimen. (**A**) is ×5 magnification and (**B**) is ×10 magnification.

**Figure 12 sensors-21-04351-f012:**
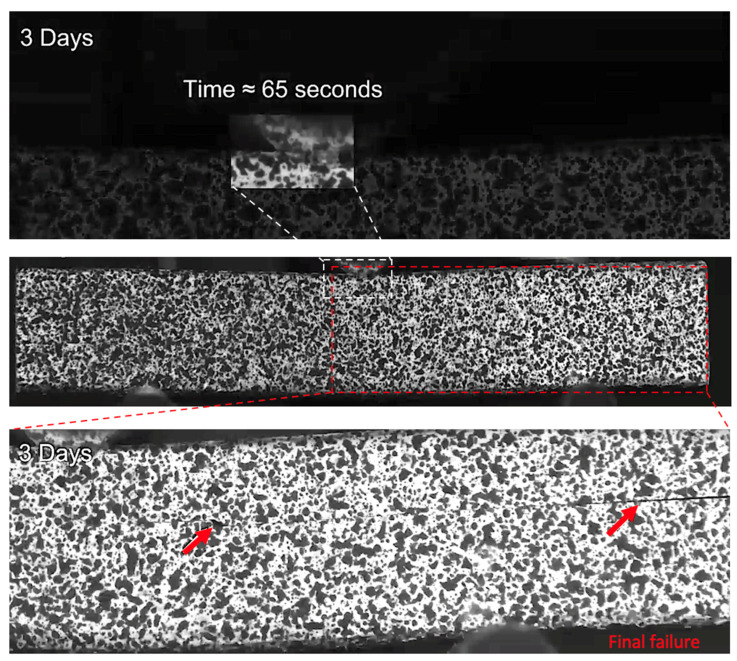
The damage observed for a 3-day water-immersed specimen prior to final fracture. The final fracture is also seen.

**Figure 13 sensors-21-04351-f013:**
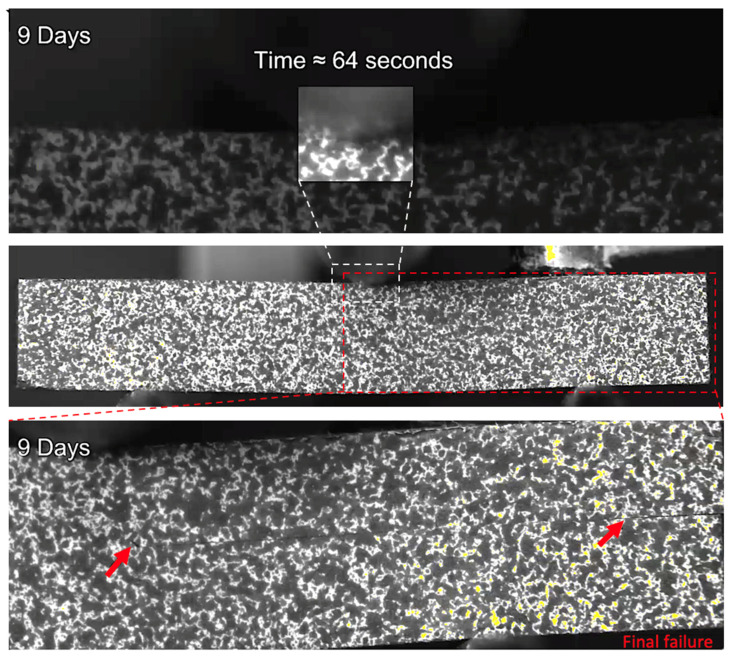
The damage observed for a 9-day water-immersed specimen prior to final fracture. The final fracture is also seen.

**Figure 14 sensors-21-04351-f014:**
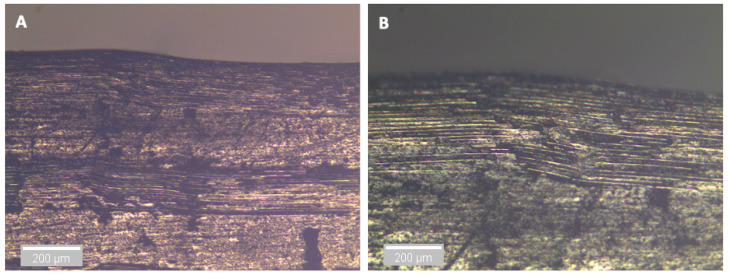
Optical microscopy of the region in contact with the loading nose for a 3-day water-immersed specimen. (**A**) is ×5 magnification and (**B**) is ×10 magnification.

**Figure 15 sensors-21-04351-f015:**
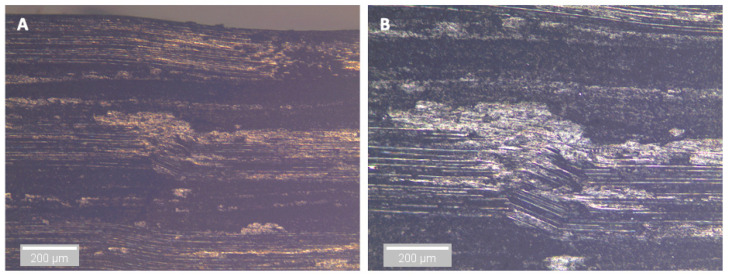
Optical microscopy of the region in contact with the loading nose for a 9-day water-immersed specimen. (**A**) is ×5 magnification and (**B**) is ×10 magnification.

**Figure 16 sensors-21-04351-f016:**
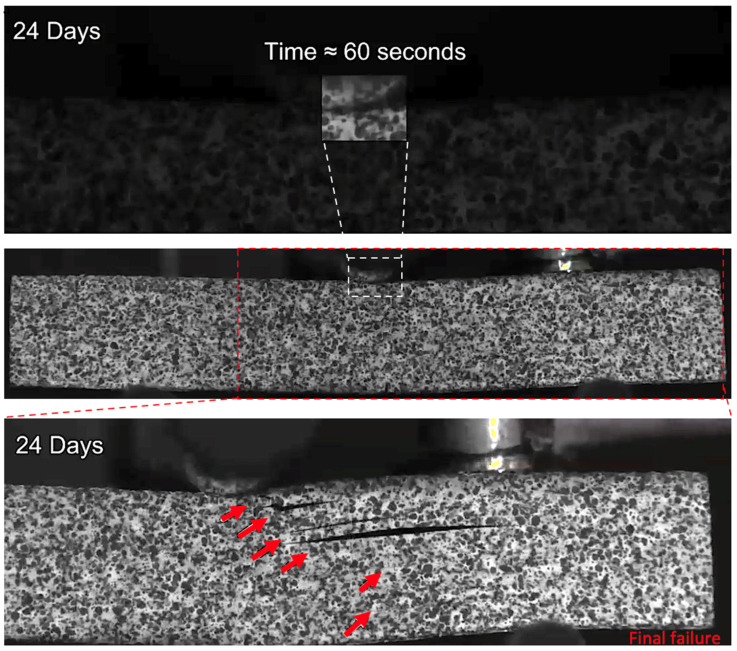
The damage observed for a 24-day water-immersed specimen prior to final fracture. The final fracture is also seen.

**Figure 17 sensors-21-04351-f017:**
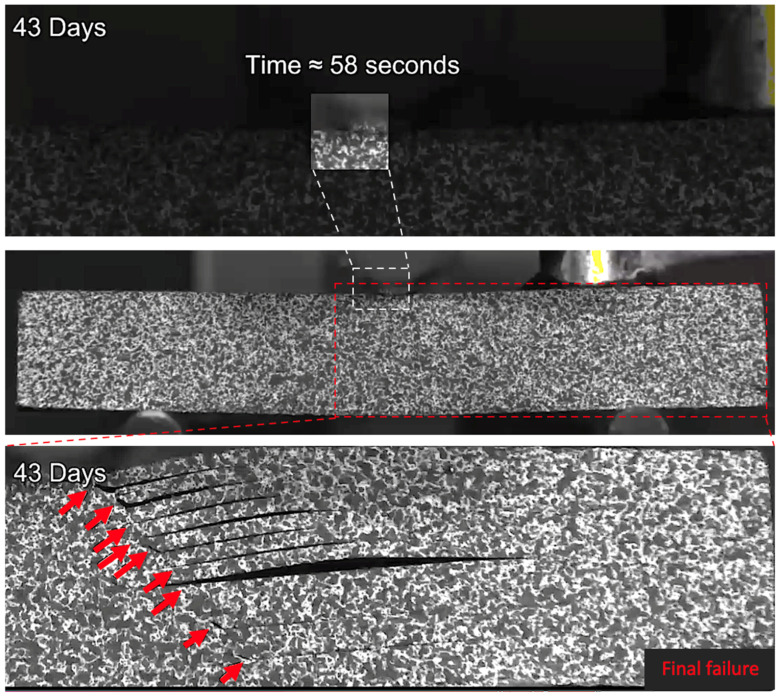
The damage observed for a 43-day water-immersed specimen prior to final fracture. The final fracture is also seen.

**Figure 18 sensors-21-04351-f018:**
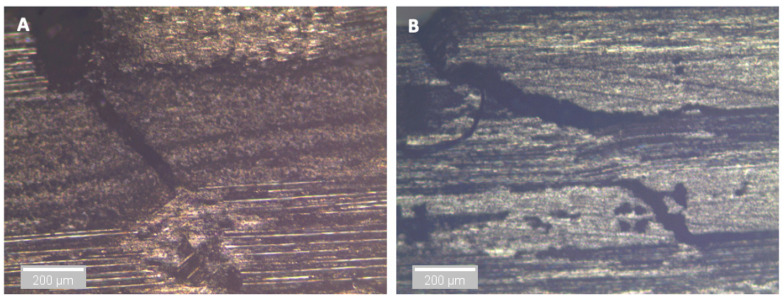
Optical microscopy of the region in contact with the loading nose for a 24-day water-immersed specimen. (**A**) is ×5 magnification and (**B**) is ×10 magnification.

**Figure 19 sensors-21-04351-f019:**
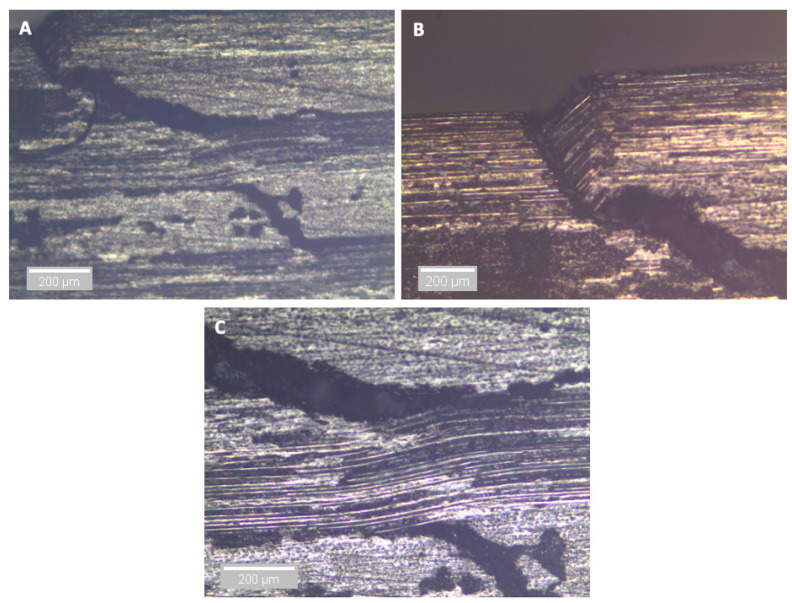
Optical microscopy of the region in contact with the loading nose for a 43-day water-immersed specimen. (**A**) is ×5 magnification, (**B**) is ×10 magnification (surface in contact with the loading nose) and (**C**) is ×10 magnification (into the second 0° ply from the surface).

**Table 1 sensors-21-04351-t001:** Correlation between the physical damage and the fracture mechanism detected with the AE technique [12].

Duration	Amplitude		
	Low (35–40 dB)	Medium (40–80 dB)	Large (80–100 dB)
Low (<1000 µs)	Class 0 (Onset matrix microcrack)	Class 1 (Propagation of matrix microcrack)	-
Medium (between 1000 and 10,000 µs)	-	Class 2 (Interfacial adhesion failure)	Class 3 (Fibre breakage)
Large (˃10,000 µs)	-	-	Class 4 (Propagation of macrocrack associated to delamination)

**Table 2 sensors-21-04351-t002:** Test matrix for specimens with 36 mm (length) by 12 mm (width).

Specimen Condition	Number of Specimens
Control (dry)	5
3-day immersion	5
9-day immersion	5
24-day immersion	5
43-day immersion	5

## Data Availability

Not applicable.

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
