# Peer review of "An Assessment of the Effect of Progressive Water Absorption on the Interlaminar Strength of Unidirectional Carbon/Epoxy Composites Using Acoustic Emission"

_sensors, 2021, doi:10.3390/s21134351_

Round 1

Reviewer 1 Report

This is an interesting topic presented in a well organized paper. The authors approach the research problem by a well established monitoring technique. However, the use of well defined NDT methods for water ingress monitoring - detection is not been mentioned by the authors in the introduction - not referenced. For instance, infrared thermography who has been used from other authors but also from the aviation industry is not being mentioned at all. One more thing that was not clearly stated in the paper is the resonant frequency used. Finally, the authors would need to provide some further info about the analysis using the OmniMet software, especially about the trimming of the area(s) of interest.

Reviewer 2 Report

This manuscript presents studies on the evolution of damage mechanisms in Carbon Fibre Reinforced Polymer(CFRP)material following water immersion for increasing durations. CFRP specimens were immersed in purified water and kept at a constant temperature of 90°C for 3, 9, 24 and 43 days. The resulting interlaminar strength was investigated through short-beam strength (SBS) testing. Non-destructive tests for unaged and aged specimens were performed using acoustic emission techniques.AE results were validated using optical measurements and microscopy. Overall, the conclusions are correct but the reviewer still has some doubts on the novelty of the current work, which should be more clearly presented in the revised version, if a revision is required. Moreover, there are also some minor errors. Therefore, the authors are suggested to improve their work according to the following comments.

Comment#1: In Abstract, the descriptions of the “Carbon Fibre Reinforced Polymer(CFRP)” in the line 14 are inappropriate, it has been used the abbreviations in the line 8. Additionally, in Introduction,it cannot be abbreviated directly when the“ CFRP ”is first appeared in the body of the article.

Comment#2: In Keywords, there is a clear error in "keyword Water absorption" in line 23.

Comment#3: In the equation  (1) (2) ,parameters are named, whereas are inconsistent with the naming in the text.

Comment#3: In the section of 2.3, the authors addressed “there are 5 specimens of each case in the SBS test”. For the curves of figure 3、6、7、8 and 9, how many samples authors used? Authors should pay more attentions to clarify these details.

Comment#4: In Paragraphs 6 of Section Results and Discussions, authors mentioned that the relationship between the amplitude and duration of the AE signals and the physical damage of CFRP taken from the study by Pérez-Pacheco et al, were used in this work, however, that study was conducted on tensile specimens, whereas the tests here were conducted under bending (SBS), complex failure mechanism is expected under shear tests. Therefore,is this assumption reasonable? How to ensure the accuracy of these experimental results?

Comment#5: The absorbed water can cause reversible and/or irreversible changes in the physical and chemical properties of the material because polymers in composite materials chemically react with water, what is the reaction mechanism? The author should clarify in the result discussion section.

Comment#6: Clear magnification should be marked on all the optical microscope images,for instance,figure11、14、15、18and19.

Comment#7: For the graphic abstract, authors should provide pictures with high resolution, which can lead reviewers and readers to understand the novelties more directly.

Comment#8: Language needs to be improved.

Reviewer 3 Report

In this paper, the authors investigated the effects of gradual change in the status of aged polymers on the overall performance of CFRP composites after 3, 9, 24 and 43 days of water immersion, respectively. The research is important and meaningful to help to expand the understanding of the role of water absorption on fibrous/polymetric structures. The paper is well written and pleasant to read. The study contains elements of novelty and can be considered for publication provided that the following points are addressed:

  1. In Figure 3, the authors demonstrate that the reduction in SBS is due to plasticization of the matrix, along with the microcracking and interfacial degradation, induced by the moisture absorption process in the aged specimens. But, two reasons, including the polymer degeneration such as the microcracking, and Young’s modulus changes caused by the moisture absorption, can both result the SBS reduction, and how to distinguish them if possible?
  2. In the short beam test, the support rollers of 3mm is smaller than the loading nose of 6mm, but it is also impossible to induce the crushing behavior and microcracking for the test specimen. So, why only the contact interface region of loading nose was investigated and discussed.
  3. Why the microcracking location presented in the Figures are not symmetrical, if the loading nose was applied on the center of the specimen. If it is influenced by the AE sensor which induces the imbalance of the actual applied load.
  4. In this paper, the three-point bend test was conducted for the short-beam strength testing, and what are the differences for the analysis results when comparing with the tension test? If they can share the similar conclusion?
  5. Some format error should be addressed, for example, the temperature degree symbol ℃ in section 2.2, the half-hidden number for several figures (Figure 1-4, 10, 11, 15), the expression problem for sentences on line 497, in which, “images taken from the from the VSG…”. The authors should correct them carefully.
